# Sustainable Agriculture: Rare-Actinomycetes to the Rescue

**Oghoye P. Oyedoh [1], Wei Yang [2], Dharumadurai Dhanasekaran [3] , Gustavo Santoyo [4] , Bernard R. Glick [5] and Olubukola O. Babalola [1,***

[1] Food Security and Safety Focus Area, Faculty of Natural and Agricultural Sciences, North-West University, Private Bag X2046, Mmabatho 2735, South Africa

[2] Institute of Environment and Sustainable Development in Agriculture, Chinese Academy of Agricultural Sciences, Beijing 100081, China

[3] Department of Microbiology, School of Life Sciences, Bharathidasan University, Tiruchirappalli 620 024, Tamil Nadu, India

[4] Instituto de Investigaciones Químico-Biológicas, Universidad Michoacana de San Nicolás de Hidalgo, Morelia 58030, Michoacán, Mexico

[5] Department of Biology, University of Waterloo, Waterloo, ON N2L 3G1, Canada

***** Correspondence: olubukola.babalola@nwu.ac.za; Tel.: +27-1-8389-2568

**Abstract:** The failure of sustainable and agricultural intensifications in saving the ecosystem/public health has caused a paradigm shift to microbiome resource engineering through sustainable approaches. As agricultural intensification systems prioritize synthetic input applications over environmental health, sustainable intensification fails to define the end point of intensification, giving room for the application of "intensification" over "sustainability" to suit farmers' needs. However, sustainable agricultural practices through microbiome resource services have been well harnessed and appreciated for their significant role in plant health and disease management due to their ability to secret agroactive metabolites with notable functionalities in a cooperative manner or as bioinoculants. The complexity of a cooperative microbiome and the uncontrollable nature of its numerous influencing parameters as well as the non-specificity associated with bioinoculant application, results in the direct utilization of agroactive compounds to obtain greater preventive efficiency. In this regard, the known bacterial trove has been seriously ransacked, yet there exists an inexhaustible bank of unknown compounds, which are conserved in Actinomycetes. However, the rare Actinomycetes group has received less attention than other plant growth-promoting bacteria; thus, the possibility exists that the Actinomycetes may encode novel useful metabolites. To unravel the possible uses of these metabolites for phytoprotection, smart culture-based techniques and genometabolomics technology have been applied. Hence the aim of this review is to express the sustainable nature of agro-antibiotics or biopesticide from these bacterial resources for the resolution of phytopathogenic havoc that reduces crop productivity.

**Keywords:** agro-antibiotics; bioeffectors; plant protection; fungicide; eco-friendly





## 1. Introduction

In 2015, the United Nations (UN) developed a vision to move the entire world into a sustainable lifestyle by 2030, which was to be preceded by the Sustainable Developmental Goals (SDG). In South Africa, the national development plan vision 2030, which is based on Stats South Africa, is 74% related to SDG's targets [1]. Therefore, sustainable practice is an all-sector-encompassing vision designed to safeguard surrounding elements such as society, economy and environment. This review focuses on how to channel this concept into the agricultural sector to resolve issues associated with food security (enshrined in SDG 2) by applying an on-farm natural resource approach of using microbes, in this case, rare Actinomycetes (also known as non-Streptomycetes), which are prolific agro-antibiotics, to assure plant health. This could help to assuage the food insecurity problems generated by the imbalance between the African population spike and available food, which put a severe

strain on the agricultural matrix, in combination with research to develop conservative measures of applying this matrix sustainably for higher productivity.

Numerous approaches for plant health assurance have been initiated over the years that prioritize ecosystem and public health sustainability, involving land rotation in the Neolithic age. However, an urban extension made land scarce for farming; therefore, agroecological principles of whole-cell inoculation were conceived as an untargeted method with minimal reduction of disease infestation, which nevertheless resulted in a stable increase in crop productivity. The desire to significantly increase productivity to meet the increasing population needs triggered agricultural intensification during the Green Revolution Era, by pumping synthetic pesticide inputs to hinder plant diseases, which met the desired outcome but came at the expense of ecosystem elements and consumers' health. From now on, the sustainable intensification principle was developed to address the drawbacks of agricultural intensification (A.I.), a method some scientists believe to be a complex decoy to escape the domain of A.I., with no defined limit of chemical inputs or effect on the environment [2–4]. Then now, are the faultless smart climate technologies that are brisker than all the alternatives and the answer to resolving the plant diseases dilemma for farmers who have the resources [5], but their combinatorial applications with intensification made this almost perfect approach somewhat problematic.

Sustainable agriculture is, therefore, the best alternative to all intensification principles as long as the natural resources of sustainability are well-managed to enhance productivity [6]. Agro-biodiversity, which is one of the natural resources manageable for sustainable production, co-exists between plants in the external environment of flowers (anthosphere), germinated seeds (spermosphere), fruits (carposphere), at the plant tissue surface (rhizoplane or phylloplane), inner tissues (endosphere) and in the root-soil vicinity (rhizosphere). In these niches, microbes interact with plants in a beneficial and complex relationship that enhances plant resilience in times of climate, human, microbial and animal-induced stress [7]. The choicest nutritious matrix—the rhizosphere—cuts across the root and soil environment, harbouring a microbial community that is shaped by the constant root exudate supply and cooperates to render below and above-the-ground ecosystem services, even when exposed to all manner of stressors. At this matrix, myriads of microbes, mostly bacterial cells, interact to enhance plant fitness and soil health [8]. Co-operational activities of rhizospheric bacteria have been studied to a significant extent, and it was discovered that they could often ameliorate many of a plant's biotic/abiotic stress factors [9–11]. For example, in a rhizobacteria study at two maize farms, it was observed that the flowering stage was dominated by *Norcardioida*, *Micromonospora* and *Frankia* species with genes that code for antagonism, siderophore producing capacity and pyoverdine phytohormones, all of which are function relevant for plant health. In addition, all of the bacterial players in this cooperative system have also been individually and synergistically applied outside their confine as bioinoculants, and their by-products have been applied as bioeffectors. *Pseudomonas* spp. and *Bacillus* spp. isolated from maize plants were reported to contain genomes that code for nitrogen fixation, phosphate solubilization, quorum sensing, trehalose synthesis, siderophores production, phenazine biosynthesis, daunorubicin secretion, acetoin synthesis, 1-aminocyclopropane-1-carboxylate deaminase activity, stress-reducing and disease control functionalities. These bioeffectors include acids/gases, enzymes, siderophores, phytohormones, enzymes, exopolysaccharides, osmolytes, volatile organic compounds and antibiotics [12]. They are more specific in action, and priorities have been given to some bacterial genera, such as *Bacillus* spp., *Pseudomonas* spp. and *Streptomyces* spp., as a source, in ensuring sustainable plant health [13], with less focus on the non-Streptomycetes class. For example, rhizobacteria (*Bacillus* and *Pseudomonas* strains) isolated from maize and soybean have been reported to elicit auxin, hydrogen cyanide and ammonia synthesis, antifungal activity, tolerance to abiotic stressors and phosphate solubilization [14]. They promote plant health through the inactivation of virulence factors, inducing plant defense mechanisms and antibiosis, of which agro-antibiotics (also termed biopesticides) have a key role to play in all.

Without question, Actinomycetes are the greatest source of agro-antibiotics or biopesticides, notwithstanding the fact that Streptomycetes strains from the rhizosphere have been nearly exhaustively studied as direct producers or as heterologous expression chassis [13]. The short and large genome contig of Actinomycetes contains thousands of biosynthetic gene clusters (BGCs) coding for known/unknown agro-antibiotics. Therefore, if the well-studied Streptomycetes contain numerous types of biopesticides from non-proteinogenic amino acid, peptides, nucleosides/analogues, acyclic, cyclic esters, organic acids, carboxylic acid esters, lactones, macrolides, amides to other minor agro-antibiotics, there may be many more of these compounds in their reserve, and in the underexplored non-Streptomycetes. The non-Streptomycetes group, as a phytoprotective agent source, has been used for the production of spinosyn from *Saccharopolyspora spinosa* (including the spinosad and spinetoram derivatives), *Saccharopolyspora pogona* NRRL 30141 derived pogonins, *Amycolatopsis* sp. derived amidenin and dehydrosinefungin derived from *Micromonospora* sp. A87-16806 [15]. Hence, more reliable novel agro-antibiotics could be produced from this trove and applied in agricultural systems to non-detrimental boost productivity.

Recent high throughput innovations in agro-antibiotics discovery involving automated culture-independent, genomics datasets and large-scale mass spectrometry-based comparative metabolomics have been used to decrypt the conserved BGCs and accurately predict modifications as well as the exact scaffold structure. These molecular networks demonstrate bioproduct furnishing encoded by orphan BGCs in the microbiome, which have to be cloned (a highly complex and technical procedure) before product release. However, the complexity and cost-ineffective nature of these molecular networks make it necessary to settle for a non-targeted culture-dependent genometabolomics roadmap [16], as highlighted in Figure 1.

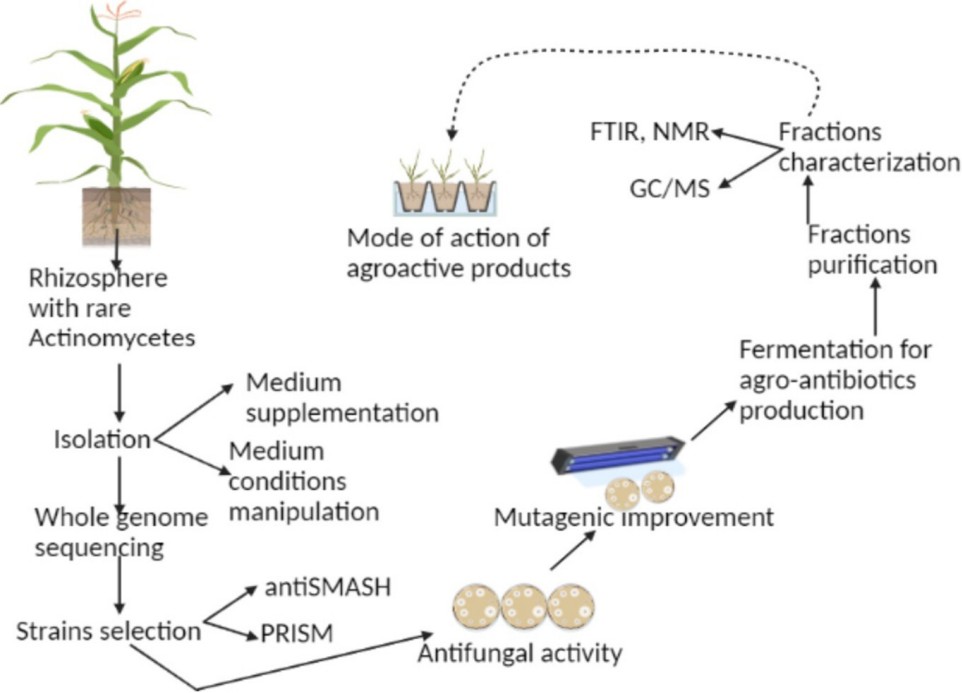

**Figure 1.** Isolation of agroactive metabolites.

## 2. History of Sustainable Agricultural Practices

Agricultural practices began from approximately 196,000 years ago, when our ancestors in the Neolithic revolution survived by consuming wild animals, plants and fungi (hence they were termed the nomadic hunter-food gatherers), to ~11,000 BCE as they adopted crop cultivation as well as rearing animals for food. Subsequently, in ~5000 BC, more focus was given to crop cultivation than to animal husbandry [17–19] and these

ancestors began to own land. They became sedentary in villages/towns close to their fields, where they adopted diverse agricultural practices such as irrigation and tillage to improve productivity and meet their population's needs. Agricultural development followed a unique track, and small farm holders experienced poor soil quality and a high disease burden. Adaptation to these challenges drove new agricultural practices, such as land rotation using the slash and burn principles. Over time, farmers were constrained to cultivating the same fields season after season [2].

As urban expansion continues to increase worldwide, land prices increase and farmland disappears. These changes provide a strong incentive to develop plants that are climate and disease resilient as well as to increase plant productivity to match the growing global population [20]. These ecosystem needs are partially addressed by integrating the right farm practices that enhance the interaction of living and non-living entities for adequate productivity with the concomitant provision of nutritious and safe food and environmental integrity, as well as food sovereignty [4]. This ecosystem approach is a sustainable one that involves recycling beneficial natural resources, decreasing superficial input utilization and diversification, as well as the amalgamation of farming systems that can build climate and disease resilience [20]. The specific practices addressed so far include mixed cultivars, intercropping, crop rotation, minimum tillage, groundwater table filling, different planting times, use of diverse varieties/species formed from conventional breeding coupled with participatory varietal selection [21], animal diversification, climate forecasting, eco-friendly pest/disease control through neem oil-based insecticide creation, compost enriched with *Trichoderma*, seed selection/coating to prevent pest infestation, then application of biogas slurry, mulching, green manure and liquid enriched composting to reduce crop loss. These strategies are adopted by subsistence smallholder farmers to enhance biodiversity management for efficient carbon sequestration, irrespective of the climatic condition or level of pest/disease infestation, in a cost and eco-friendly manner [20]. This is aimed at ensuring that all beneficial entities that stimulate plant productivity are enhanced; however, some of the practices, such as animal diversification, still promote faecal pathogens seepage into the soil.

Scientists have recently searched for means of improving crop productivity in an effort to match the persistently increasing population; this initially led to the creation of the Green Revolution, involving the utilization of synthetic fertilizers/pesticides that increased productivity at the expense of soil quality, biodiversity, food consumer's health and the environment [22], these detrimental effects triggered the S.I. approach, which some scientists see as a complex illusion that fails to delineate the extent of intensification. The S.I. approach is cost ineffective, depletes natural resources, and has no definite limit of chemical input application, and hence has a resultant undefined level of environmental impact. It is a controversial concept that did not clarify which is more important, intensification and its provision of adverse environmental impacts or sustainability with its excessive land use [3].

With the constant increase in a global and commensurate upsurge in food insecurity, an artificial intelligence approach to agricultural development was embodied with smart multi-faced technologies. The major trade-off of these technologies is that they must be combined with an intensification input; for example, agricultural drones, having conducted a speedy/thorough health assessment, must apply synthetic pesticide or fertilizer as a corrective measure [5], which alters the sustainability of the holistic concept. Therefore, this yet-to-be-debated, promising tech-agricultural innovation in the twenty-first century has to be supported by a more sustainable, potent and affordable alternative agent/input that could be applied to alleviate plant stress, which is the major cause of soil infertility as well as food insecurity. As an alternative to synthetic chemical applications, the application of plant growth-promoting bacteria as a direct and indirect means to sustainably increase productivity has been initiated. Under this premise, bacterium/bacteria were pulled from a plant/soil site and utilized for crop growth promotion in another site as phytoenhancers or phytoprotectants. In addition, Actinomycetes are excellent producers of natural products due to highly conserved genome encoding metabolites in thousand contigs, some of which

have been mostly exploited for indirect plant growth promotion through plant protection during biotic stress [23]. Also, considering the history of Actinomycetes as the biggest producers of antibiotics, which has greatly exposed their antimicrobial potential in killing or inhibiting microbes [24], some of the products have been diverted for sustainable agricultural development because they are mostly predominant in the soil as defensive or offensive molecules, and their encoding genes are described in Table 1.

**Table 1.** Roles of rare Actinomycetes in sustainable agriculture.

| Rare Actinomycetes | Core Genes | Metabolites Encoded | Functionalities | Roles in Sustainable Agriculture | References |
|---|---|---|---|---|---|
| *Frankia* sp. | *nif* | Nitrogenase enzyme | Nitrogen fixation | Soil fertilization | [25] |
| *Tsukamurella tyrosinosolvens* | *pho* | Phosphatase enzyme | Phosphate solubilization | Phosphate fertilization | [26] |
| *Amycolatopsis* sp. | $PR_{1-1a}$ & *GLU* | proteins | Systemic acquired resistance | Plant defence | [27] |
| *Arthrobacter* sp. SD3-25 | *atzB, atzC* & *trzN* | hydrolases | Atrazine & simazine pesticide biodegradation | Soil fertility and Bioaugmentation | [28] |
| | *mbtH, fagD* | protein | Siderophores biosynthesis | Plant defence and Iron fertilization | [29] |
| *Tsukamurella tyrosinosolvens* | *febB, febD, yqjH, hpaC* | Transport proteins | Biosynthesis of iron transporter | Plant defence and growth | [26] |
| | *atzF* <br> *argG* <br> *argH* | Hydrolase <br> Arginine succinate synthase <br> Arginine succinate lyase | Urea degradation | Ammonia and amino acids biosynthesis for growth | [29] |
| *Saccharothrix* sp. | *SacA,B,C,E* | Polyketide synthase | *Saccharochelins* A–E biosynthesis | Phytoprotection | [30] |
| *Amycolatopsis* sp. | *asrR* | Type III glycopeptide | Ristomycin | Phytoprotection | [31] |
| Saccharothrix yanglingensis Hhs.015 | *Chi6769* | Protein | Chitinase biosynthesis | Phytoprotection | [32] |
| *Rhodococcus ruber* $C_1$ | *dmpP* | Phenol hydroxylase | Phenol degradation | Biodegradation | [33] |
| *Rhodococcus* sp. ANT_H53B | *crtP,M,N,Nc* | Diapolycopene Oxygenase, dehydrosqualene synthase & other enzymes | $C_3$ apocarotenoid biosynthesis | Phytoprotection and enhancement | [34] |
| *Nonomuraea* sp. NJM5123 | NRPS gene *ecuE* | Enzymes that form tridecapeptide full length | Ecumicins biosynthesis | Antuberculosis activity | [35] |
| *Nocardia vaccinii* NBRC15922 | *AuaJ* | Epoxidase *LacC* | Lasalocid | Antibacterial activity | [36] |
| *Saccharopolyspora* sp. | *arsG* | Arsenate reductase | Arsenic removal | Arsenic biocleansing | [37] |

## 3. Sustainable Agriculture in the Resolution of Problems in Africa

On a worldwide scale, out of ~570 million farms, 513 million are estimated to be from smallholders, tracking down to Africa, where food insecurity, poverty and hunger

is severe, 80% of the food supply also comes from small farms. This makes smallholders pivotal in meeting the vision enshrined in SDG 2, aiming to improve nutrition, achieve food security and promote sustainable agriculture [38]. Most of their farmlands depend on seasonal climate for productivity [2], and the problems caused by global warming have been reported to be more rapid in the African continent, with predominant variations in the north and south of arid regions than in humid central Africa [39]. These climatic conditions, which most of the smallholders depend on for agricultural production, resulting in reduced access to water, shorter growing seasons, longer water deficit periods due to irregular climate, higher evapotranspiration, accelerated land degradation due to high temperature and diversity in the distribution of phytopathogens [40]. Unavoidably, the resulting low productivity, loss in soil fertility and poor plant health deters the projected plan to improve crop production to 60–100% to meet the population by 2050. In short, the major problems associated with agriculture in Africa can be resolved by enhancing climate and disease resilience in small farms for stable productivity. A sustainable approach for the establishment of disease resilience is the application of rhizosphere soil biodiversity, which has been reported to ensure plant resilience to stressors in plants, especially in arid zones [41]. Rhizo-soil biodiversity enterprises for increasing disease resilience range from cooperative management in plants and whole-cell application as a bioinoculant to collective strains utilization for plant growth promotion and plant protection [23,42]. The cooperative management of microbiomes in dealing with abiotic stressors has been well-studied for maize plants through plant growth promotion and has been currently revisited [9,43,44]. In addition, the synergistic application of *Pseudomonas* sp. and *Bacillus* sp. in plant rhizosphere for plant health has been reported to produce useful metabolites [45,46]. *Streptomyces* sp. has also been used as a bioinoculant to resolve microbial phytopathogenic problems in rhizosphere soil and ensure abiotic stress resilience [14]. The whole cell/bioinoculant approach is non-specific, but one of the direct approaches to harnessing natural resources for plant protection is the use of biopesticides or agro-antibiotics. In recent times, biopesticide studies have gained attention in agricultural development considering the recalcitrant nature of synthetic pesticides and the excellence of *Bacillus thuringiensis*, which has driven new product creation such as avermectin from *Streptomyces avermitilis* and the multi-site active spinosad (a mixture of spinosyn A & D) derived from *Saccharopolyspora spinosa*. Given the latter, Actinomycetes provide a unique possibility for potent, safe and site-specific bioagents for plant protection, as they are well endowed with potent gene clusters [47].

This unique shift from bioinoculant application to the engineering of their active agents for sustainable agricultural development was conceived based on the multifunctional roles displayed by rare Actinomycetes as a phytoenhancer (in ensuring plant growth promotion) and a phytoprotectant.

### 3.1. Rare Actinomycetes as a Phytoenhancer

Rare Actinomycetes can maintain soil fertility by fixing the macronutrients needed for plant growth and development. Rare Actinomycetes such as *Frankia* spp. have been reported to fix atmospheric nitrogen in leguminous and non-leguminous plants by biosynthesizing nitrogen through atmospheric nitrogen conversion to nitrate or ammonium ions. They are the top players in nitrogen fixation, as they are more metabolically active than customary nitrogen-fixing *Rhizobium* sp. The rare Actinomycete–plant relationship has provided plants with usable nitrogen that is absolutely adequate for chlorophyll pigment, nucleic acid, amino acids, proteins and energy (in the form of ATP) biosynthesis [25]. In addition, *Frankia* sp. var. A & B induced plant nodulation (with variant A revealing more vesicles within infected plant cells than variant S) when inoculated into *Alnus glutinosa* [48]. In addition, through phosphatase/phytase enzymatic activities as well as organic acid secretions, rare Actinomycetes can solubilize the mass deposit of insoluble phosphorus in the soil to soluble phosphorus, hence enhancing plant respiratory chain components, energy transduction processes and most macromolecules synthesis [49]. A typical example is the *Tsukamurella tyrosinosolvens* $P_9$ strain isolated from tea plant soil rhizosphere, which

promoted plant seedling growth in a pot experiment through organic acid chelation [26]. *Micromonospora* and *Nocardia* strains have also been reported to dissolve potassium-based minerals in the soil through organic acid chelation for potassium ion release to plants. This state of potassium is required by plants for regulation of stomata pores, which open for plant detoxification (in times of heavy metal or other toxicant stress) or close to prevent water loss during drought stress [25], hence inducing plant systemic resistance to abiotic/pollutants stress as well as reviving soil fertility. Aside from macronutrients, plants also require micronutrients such as zinc, cobalt, iron and chromium in millimolar concentrations for proper metabolism. Concentrations exceeding the limit required could deform chromosomal conformity, and Actinomycetes have been reported to modulate these minerals during their accumulation or sequester such metals during a shortage in plants. A typical example is *Nocardia* spp., which are engaged in the bioremoval of heavy metals of diverse types [50]. Genera such as *Corynebacterium*, *Rhodococcus*, *Gordonae* and *Mycobacterium* have been reported to remove hydrocarbon and other environmental toxicants due to their high tolerance to extreme conditions and massive catabolic activities owing to the luxuriant enzymes they possess. They apply various enzymes such as reductase, peroxidase, laccase, ligninase, chitinase and chitosanase for the biodegradation of organic toxicants [51]. Rare Actinomycetes enzymes play a major role in carbon cycling through the conversion of humic substances to agroactive metabolites released into the soil which induce systemic resistance to plant stress and enhance the soil fertility and nutrient fertilization of plants [52]. Saccharothrix sp. D09 enhance plant root growth during iron limitation in plants by chelating $Fe^{3+}$ into $Fe^{2+}$, which is needed for plant development. Fe chelation is successful through siderophore (saccharochelin A–E) iron complex formation and subsequent $Fe^{2+}$ dislodgement into the plant environment. Siderophores, phytohormone (indole acetic acid) and ammonia produced by *Micromonospora aurangtinigia* alleviate salt stress by enhancing plant root and shoot growth [53].

### 3.2. Rare Actinomycetes as Phytoprotectant

Generally, Actinomycetes' role as the chief bioagents or biocontrol agents or phytoprotectant agents in agricultural production is indisputable. Actinomycetes as well as other rhizobacteria can release antibiotics, volatile organic compounds, hydrogen cyanide, lytic enzymes and siderophores in plants during biotic stress through a process called antibiosis, to inhibit or kill the phytopathogens that result to low crop yield. In particular, Actinomycetes are a prolific inexhaustible supplier of antibiotics applied against pathogens of clinical and agricultural origin [54]. Although they have been explored for other phytoprotective metabolite production, such as volatile organic compounds, hydrogen cyanide and siderophore production, they are always the first option for the production of antibiotics relevant for plant protection against a wide spectrum of pathogens. In addition, rare Actinomycetes have been reported to induce systemic resistance and acquired resistance by increased antioxidant enzyme secretions and the expression of LOX/APX transcriptions, as well as increase in the expression of $PR_{1-1a}$ and GLU genes. These gene expressions have enhanced cucumber fruit yield under greenhouse conditions without infection [27]. Some phytohormones, volatile organic compounds, antibiotics and siderophores have also been reported to induce systemic resistance in host plants. In addition, rhizospheric soil-derived actinobacterial species displayed antagonistic activity against *Fusarium oxysporum* by producing hydrogen cyanide, hydroxamate and indole acetic acid, which hindered the growth of the potent wilt pathogen in *Gladiolus* sp. Multifunctional phosphorus solubilizing *T. tyrosinosolvens* significantly promoted peanut seedling growth by secreting hydrogen cyanide and siderophores [26,55]. *Nocardiopsis alba* has been reported to protect palm oil plants from the aggressive pathogenic fungus *Ganoderma boninense* by secreting volatile organic compounds such as 5-methyl ethanethioate, 1,2-dimethyl disuphane, acetic acid, 2-methyl propanoic acid, 3-methyl butanoic acid, nonan-2-one and 2-isopropyl-5-methyl cyclohexan-1-ol [56].

Furthermore, within the rhizosphere of plants, the predominance of Actinomycetes in combating diseases has been extensively studied and their role as bioinoculants owning to the metabolites they possess [23]. Hence, the focus of this review is on leveraging the antibiotic metabolite as an active agent (in this case, agro-antibiotic or biopesticide) for plant protection.

## 4. Rare Actinomycetes to the Rescue

Actinomycetes and/or their agroactive innovations are a safe method of ameliorating biotic stress by producing chemicals with disease-suppressive activity in diverse plant structures without distorting the external environment, ensuring rapid biodegradation (with co-current optimization of nutrient use efficiency), providing no possible genetic transfer, delivering all soil and plant ecological enterprise as well as creating rhizosphere microbial community balance through their natural selectivity [47]. Actinomycetes have been reported to produce ~10,000 (i.e., 45%) of ~23,000 metabolites released by microorganisms, and 20–30% of the 10,000 compounds are derived from non-Streptomycetes or rare Actinomycetes; hence, these groups have been underutilized. Consequently, the attention of researchers is now directed towards the use of non-Streptomycetes or rare Actinomycetes such as *Micromonospora*, *Streptosporangium*, *Microbispora*/*Streptoverticillium*, *Saccharomonospora* and *Norcadia* species as crop protective tools (based on their antibiosis and growth promotion activities); other genera include *Actinokineospora*, *Actinomadura*, *Allostreptomyces*, *Amycolatopsis*, *Dactylosporangium*, *Kutzneria*, *Lechevalieria*, *Microbacterium* species [54], with 70–80% yet to be identified. A metagenomics study was conducted on mineral soils in a dry valley, with the discovery of multiple uncultured *Pseudonocardia* and *Nocardioides* species. In addition, several agroactive effectors have been commercially produced from *Streptomyces* sp., with a large spectrum of action against plant diseases, and these treatments are reported to reduce plant disease severity by 95%. These microbial-based agroactive effectors have also been reported to act as enhancers of nutritional and functional quality as well as to increase the yields of fruits and vegetables [57]. These effectors can be used directly in a sustainable manner, with the benefits shown schematically in Figure 2, to control the upsurge in global plant pest/disease burden and enhance crop productivity.

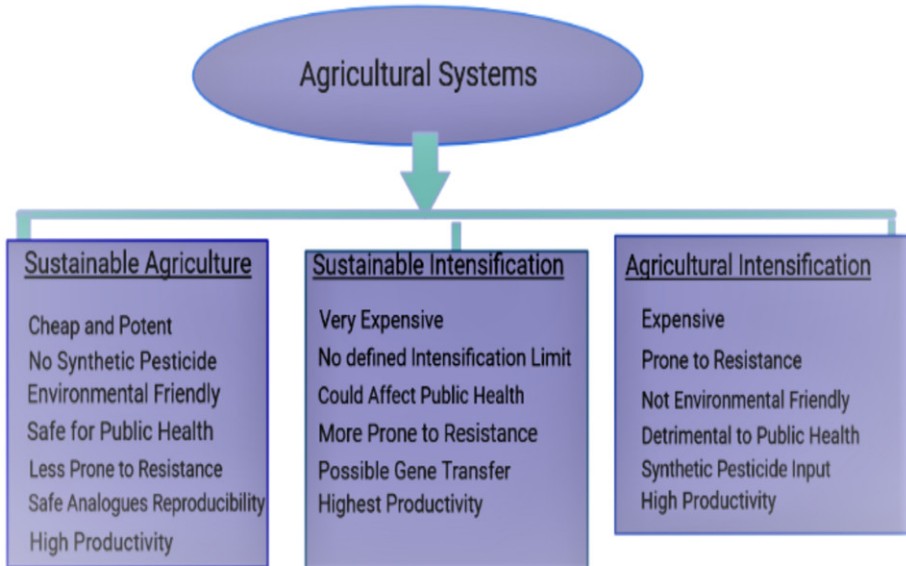

**Figure 2.** Characteristics of agricultural systems applied for ensuring disease resilience.

## 5. Agro-Antibiotics Encoded by Biosynthetic Gene Clusters (BGCs) in Actinomycetes

Actinomycetes are Gram-positive, filamentous cocci, prokaryotic organisms that are 1–2 μm in diameter and respire both aerobically and anaerobically. They are abundant in

many soils (often $10^6$–$10^8$ cells/g), marine environments (5–40 Cfu/mL) and desert soil habitats and are known producers of a large number of different metabolites. Actinomycetes are a hub of known and unknown bio-products.

The biosynthetic gene cluster sequences encode diverse proteinic and non-proteinic enzymes disbursed in modular frames forming diverse BGC types, which include: PKSs (polyketide synthetases), NRPS (non-ribosomal peptides synthetases) and hybrid-PKS-NRPS in assembly lines. Most of the bio-products for crop protection, such as avermectin and its derivatives, milbemycin, meilingmycin, spinosyn and its derivatives, pogonin, polynactin, herboxidiene, sannastatin, coelimycin P1, actinorhodin, rubiginone D2, juniperolide A, tetramycin, antimycin, nemadectin and actinospene, among others, are synthesized from PKS assembling lines, with a few NRPS and hybrid derived products [58]. Hybrid and NRPS-derived phytoprotective bioproducts are yet to be produced, mostly silent under in vitro studies, requiring planta discovery with metagenomics accompanied heterologous expression of their biosynthetic genes in suitable chassis for reproducibility [59]. The gene cluster sequences encode diverse modules of complex PKSs (polyketide synthetases), NRPS (non-ribosomal peptides synthetases) and hybrid-(i.e. PKS-NRPS) in assembly lines. These enzymes aid the transfer of a series of monomer units to appropriate linear oligomers until the formation of a known or orphaned product [60]. The biosynthetic genes consist of core codifying genes and regulatory long operon/promoter genes that either inhibit or promote core gene expression under the influence of a changing environment. The genes of PKS, NRPS and hybrid assembly lines have been characterized, which confirmed that such genetic design is in clusters that should facilitate their complete pathway cloning [61].

In fact, several studies have revealed the cloning of multiple copies of biosynthetic genes to optimize metabolite production, the swapping of PKS genes of closely related species or classes to enhance the spectrum of metabolites produced and the shuffling of genes in diverse rounds through the physical/chemical mutation of Actinomycete genomes to ensure phytoprotection. These manipulations have led to the innovation of new derivatives of avermectin, spinosyn and novel erythromycin [47]. Metabolites characterized by these organisms have been used in pharmaceutical industries as antibiotics as well as in agricultural sectors as biopesticides or agro-antibiotics. Agro-antibiotic compounds are produced in low doses by a microorganism against other organisms; they are specific in action, self-propagating, less prone to resistance, compatible with irrigation activities, mostly biodegradable, have preventive or killing control and are not crop-specific, and the producer possesses a transgene that makes the producing organism resistant to the by-product they exude. From a site-specific agro-antibiotic, other derivatives with a wider spectrum of action can be obtained by genetic or synthetic modifications. For example, synergistic activity is seen in the combination of spinosyn A and D to form spinosad with a wider spectrum of action against thrips, armyworms, codling moths, cutworms, leafminers, mosquitoes, ants, fruit flies, spider mites and other varieties of pests [62]. Numerous compounds or formulations have been derived from *Bacillus*, *Pseudomonas* and *Trichoderma* species [63]. Presently, the global market for agro-antibiotics is expected to grow to USD 4.5 billion, with 60% of the compounds coming from Actinomycetes (Streptomycetes).

It is worth noting that Actinomycetes have a large number of BGCs that are uncharacterized or conserved; the genome is made up of repeats, functional domains and genome assembly lines connected to make diverse metabolites scaffold [64]. Hence, studies have revealed the presence of large numbers of cryptic compounds in Actinomycetes, which could only be decoded if constantly characterized; perhaps this is the reason why it seems Streptomycetes compounds encode more than their non-Streptomycete counterparts. For instance, a study conducted on *Streptomyces*-derived biopesticides reflected more than 100 compounds from about twelve classes of bioherbicides [15]. It is, therefore, imperative to decrypt these silent metabolites to resolve disease burden in a more sustainable manner, focusing on non-Streptomycetes.

Furthermore, outside the PKS, NRPS and hybrid pathways that are chiefly responsible for secondary metabolite production at the stationary or death phase of Actinomycetes



growth, other pathways that are implicated in biosynthesis include β-lactam, oligosaccharides synthesis and Shikimate pathways [65]. Biopesticide classifications based on target pest are grouped into acaricides (target mites), fungicides (target fungal specific processes), bactericides (target bacterial specific processes), herbicides (target plant-specific processes), larvicides (affect insect larva), nematicides (affects roundworms), termiticides (target termites) and ovicides (affect insect eggs). However, based on existing agro-antibiotics, they can be grouped basedon their agroactive efficacy, as elaborated below.

### 5.1. Herbicidal Agents

Weeds have globally resulted in an estimated 34% annual loss in crop yield through their adverse competition for crop resources; bioherbicide application is the best eco-friendly remedy. Bioherbicides are weed target compounds of less than 500 molecular weight sizes with low toxicity in active doses to the ecosystem and have enhanced their relevance as a biointensive integrated weed control agents for agricultural development. These agents can be peptides, amino acids, organic acids, amides, carboxylic acid esters, lactones, macrolides, nucleosides and analogues that inhibit root elongation and ethylene synthesis; they can inhibit photosynthesis systems [66], root and shoot elongation, amino acid, cellulose, glycan, nucleic acid and de novo fatty acid biosynthesis in plants. Large numbers of Streptomycete herbicides have been derived, with only sparse studies on rare Actinomycetes. Some rare Actinomycete-derived herbicides include patented maiden/derivatives from an *Amycolatopsis* sp. strain in Japan, which is used to reduce sulfonylurea herbicide application in rice farms; carbocyclic coformycin from *Saccharothrix* sp., which disrupted the ATP pool in plant cells; *Actinoplanes* sp.-derived formycin A & B with a broad spectrum of post/pre-emergence herbicidal activity against monocot weeds and grasses; thiolactomycin from *Nocardia* sp.; *Micromonospora* sp. A87-16806-derived patented dehydrosinefungin herbicides and *Actinomadura madurae*-derived ribofuranosyl triazolone, which is active against weeds and grasses [15].

### 5.2. Insecticidal/Acaricidal Agents

The environmental and public health impacts of synthetic pesticides have stimulated the development of microbially produced compounds against insects and mites. The application of agro-antibiotics such as avermectins and emamectin from rare Actinomycetes as both acaricidal/insecticidal activity operates through the activation of the nerve endings of insects and mites to extend the duration of the opening of GABA (gamma-aminobutyric acid)—gated Cl ion channel abnormally, resulting in the release of chloride ions and GABA, which highly polarizes the nerve membrane potential, blocking the electrical nerve conduction. This stimulates symptoms such as moulting, disturbance in water balance, metamorphosis and reproductive developments, with no cross-contamination [66]. Spinosyn A & D families, a 21-carbon tetracyclic macrolide produced by *Saccharopolyspora spinosa*, is an active neonicotinoid insecticide that is active against some species of Coleoptera/Orthoptera as well as Thysanoptera, Diptera and Lepidoptera, with little or no effect on non-targeted insects or mammals. Asides from acting on GABA receptors, they activate nicotinic acetylcholine receptors, which results in nervous system excitation, involuntary muscle contractions, tremor and paralysis. A second-generation spinosyn product called spinetoram (a semisynthetic natural pesticide), with a broader insecticidal spectrum with a more favourable toxicity profile in mammals and in the environment, has also been created [47], along with *Saccharopolyspora pogona* NRRL 30141-derived pogonin (21-butenyl-spinosyns), which possesses the same spectrum of action as spinetoram against sucking insects such as cotton aphids and tobacco budworms [67]. In addition, Actinomycetes enzymes chitinase can regulate the process of chitin formation in insect mites and pests by disrupting the N-acetyl-D-glucosamine chitin chain at the β-1,4-linkage between monomeric subunits; this affects insect feeding, digestion, nutrient utilization and growth, and also indirectly causes deformities [47].

### 5.3. Anti-Phytopathogenic Agents

In smallholding farms from which the major food supply is derived, insects, pathogens, weeds and mites are the main threat to food security worldwide [68]. Pre-harvest fungal pathogens of the most significant cash crops, such as maize, as well as oil crops such as sunflower, include species of *Aspergillus*, *Fusarium* and *Cercospora zeae-maydis*. In addition, *Sclerotinia sclerotiorum* is highly pathogenic towards sunflower plants and acts by causing head and stalk rot. The crop resistance to these fungi by rhizosphere microbiome activities has encouraged scientists to utilize their agroactive metabolite resources for biointensive pest management. To this end, several antifungal agro-antibiotics have been elucidated from Streptomycetes—the most studied Actinomycetes. A recent analysis of the rare Actinomycetes has identified echinosporin and 7-deoxyechinosporin from *Amycolatopsis* sp. YIM PH20520 derived from the rhizosphere of Panax notoginseng, which displayed a high level of antifungal activity against root rot pathogens of Panax notoginseng [69]. Once a fungicide-exuding strain has been selected, several innovative approaches could be applied to its productivity and efficacy. Such approaches have been utilized on the highly overhauled Streptomycetes group to enhance fungicide yield and activity and activate the expression of more silent clusters. Actinomycete biosynthetic gene clusters are so malleable that sophisticated technical approaches such as heterogeneous gene transfer, overexpression of a specific gene in a mutant strain, the deletion or disruption of insignificant genes and the replacement of PKS genes of a target compound with another gene can bring about a variety of enhancements. These strategies are highly technical and expensive to achieve; however, less expensive approaches have been applied to *Streptomyces* species with positive results. These classical approaches involve agro-antibiotics supplementations of fermentation medium as well as an untargeted physical/chemical mutagenesis approach applied to express new or more active analogues from candidate strains [54,70].

Some antimicrobial metabolites were released by *Actinomycetes*, which were active against plant viruses. For example, *Actinomycete*-derived ε-poly-lysine was active against tobacco mosaic virus by acting as a curative and protective agent. In addition, with a minimum inhibitory concentration of 0.2 μg/mL–15.6 μg/mL, *Actinomycete*-derived munumbicins A–D was potent against the bacterium *Pseudomonas syringae* [71].

### 6. Targets of Fungicide

Actinomycete-derived fungicides have diverse mechanisms of action; they can disrupt the cell membrane and/or target cytoplasmic organelles (deoxyribonucleic acid, protein, ribonucleic acid and mitochondria). Examples of fungicides derived from Actinomycetes include bafilomycin, resistomycin, netamycin, tetracenomycin D, antifungalmycin 702, filipin III, strevertenes, and oligomycins A & C, all derived from species of *Streptomyces* and 5-hydroxyl-5-methyl-2-hexonoic acid derived from *Actinoplane* sp., which displayed antifungal activity against a broad spectrum of diverse crop ravaging fungi [52].

### 6.1. Cell Membrane Target

Fungicides with small cationic polypeptides are amphiphilic, i.e., with a hydrophilic head consisting of active polar amino acids endowed with cations at the N-terminus and hydrophobic tail that is framed by acylation of amino acids at the C-terminal. They mostly target fungal cell membranes, which are composed of glycoproteins, chitin, polysaccharides and a glucan frame (made up of alpha-1,3-glucan, beta-1,3-glucan, beta-1,6-glucan and beta-1,3-/beta-1,4-glucan) that maintains the cell metabolic activities, ion exchange and osmotic pressure within a confined zone and acts as a protective circular shield. *Streptomyces lavendulae* strain X33 crude antifungal extract impaired the membrane integrity of the *Penicillium digitatum* pathogen [72]. Netamycin is a cell membrane-specific fungicide that causes the release of amino acids and electrolytes from the membranes, hence destroying membrane permeability. Fungicides specifically target glucan, chitin or lipids by insertion or attachment, binding and disruption, which eventually leads to cell death depending on the concentration/potency of the fungicidal compound [66,73]. The major

fungal membrane lipid components are sterols (ergosterols, which are responsible for maintaining cell membrane integrity), glycoglycerolipids and sphingolipids. Reduction in ergosterol formation induced by *Streptomyces* sp. strain N2-derived antifungalmycin N2 elicited cell membrane disruption in *Rhizoctonia solani*. Fungicides are electrostatically attracted to the membrane surface, and then pile up to form monomers or polymers. In the monomeric complex, the hydrophobic portion of the fungicide binds to the membrane phospholipid bilayer at the hydrophilic ends to form an ionic channel or pore [54,74]. This complex accumulates to become polymeric, forming transmembrane pores and distorting the membrane integrity as well as seriously impairing the membrane fluidity. Fungicide polypeptide chains can reframe the cell membrane molecules by exchanging the calcium ion $(Ca^{2+})$ or magnesium ion $(Mg^{2+})$ of membrane lipopolysaccharide with their hydrophilic or hydrophobic regions [73].

Chitin synthase inhibition is caused by most nucleoside-based fungicides bound to a polypeptide moiety that is structurally synonymous with UDP-N-acetylglucosamine (chitin synthase substrate) and out-competes chitin synthases due to its higher affinity for N-acetylglucosamine. This competitive displacement of chitin synthase from substrate linkage results in osmotic swelling of the fungal pathogen mycelia after a series of defects in cell wall maturation, septum/budding formation and cell division. For example, polyoxins and nikkomycins, which are similar to UDP-N-acetylglucosamine are potent antifungal agents. Some fungicides also inhibit beta-1, 3-glucan synthase, which affects the fungal membrane frame, causing ultrastructure modifications through a reduction in ergosterol [66].

*6.2. The Target of Cytoplasmic Organelles*

Having penetrated the cell membrane, fungicides can inhibit DNA-dependent RNA synthesis by engulfing RNA polymerase. Most Actinomycete-derived fungicides that affect cytoplasmic organelles belong to the Streptomycete groups. For example, blasticidin and kasugamycin, which interfere with peptidyl transfer to the ribosomal subunit, prevent protein synthesis. Streptomycin causes codon misreads, which lead to the synthesis of non-functional proteins. *Streptomyces* spp.-derived streptotricins block protein synthesis in blast causative fungi. Most fungicides also inhibit mitochondrial respiration by eliciting non-lytic discharge of ATP into the cytoplasm, where it binds purigenic receptors. For example, a crude extract obtained from *Streptomyces lavendulae* strain X33 elicited mitochondrial membrane dysfunction, which affected the energy metabolism of *Penicillium digitatum*, while other fungicides inhibit mitochondrial respiration by blocking the function of respiratory complex II & III or binding to the Qi ubiquinone binding site [66,72,74,75]. *Streptomyces* sp. strain N2-derived antifungalmycin N2 is a typical fungicide with a high affinity for the ubiquinone binding site, thus inhibiting the activity of the succinate dehydrogenase enzyme and interfering with the mitochondrial metabolism in *Rhizoctonia solani*, and lytic polynactin sponsors the leakage of cations from the mitochondria [74].

## 7. Novel Fungicide Mining through Genome Sequencing

The pressure on scientists to improve fungicide options in phytomedicine in order to reduce the rate of fungal phytopathogenic impact in the agricultural system in an eco-friendly and sustainable manner has created diverse roadmaps for the isolation and selection of competent producing strains. The culture-dependent route for the identification of bioproducts is a sure way that reflects the ecology and metabolism of the biodiscovery, even if it does not capture the actual environmental diversity, as some organisms are viable but remain in the dormant stage. This is a stage that most rare species occupy, hence enhancing the overgrowth of the r-strategist (fast-growing species) fast grower over them (which are the k-strategist slow growers). However, many smart culture-dependent systematic approaches mimicking their natural environments have been applied to isolate most microbial diversity and also reduce the rediscovery of known metabolites. For example, the co-culture procedure used to grow *Tsukamurella pulmonis* TP-B0596 alongside *Streptomyces lividans* on solid and liquid agar media stimulated the pigment-releasing capacity in

*Tsukamurella pulmonis* TP-B0596, a trait displayed by other genera (such as *Corynebacterium*, *Rhodococcus*, *Dietzia*, *Gordonia*, *Mycobacterium* and *Williamsia*) in trials against *Streptomyces lividans*. The pigment production was also accompanied by extracellular production of mycolic acid by *Tsukamurella pulmonis* TP-B0596 and the other genera tried. It was also observed in this bacteria–bacteria communication model that physical contact was required, as the culture was found to produce the pigment at the contact point on the agar. Subsequently, novel arcyriaflavin E (an indolocarbazole skeletal alkaloid) was produced through co-culture isolation of *Tsukamurella pulmonis* and *Streptomyces cinnamoneus* NBRC 13823, as well as *Tsukamurella pulmonis* and *Streptomyces* sp. CJ-5 for the production of chojalactones A–C (a novel butanolides). In other culture-dependent medium modifications involving variant temperature and nutrient application, a temperature increase of 2 °C has been reported to endogenously activate validamycin BGC in Actinomycetes. Aside from the use of the combined culture method in mimicking the natural environment, single or multiple manipulations of temperature, moisture, light intensity, pressure, oxygen content (physical conditions), nutrients availability, pH, nature of carbon source (chemical environment) and host factor (i.e., the element's host exude) are parameters that can be co-opted in the media preparation to stimulate untargeted BGCs expression. For example, sufficient carbon supplementation of the isolation medium has been reported to repress the secondary metabolite pathway and influence BGC expression. This mechanism of down regulating secondary metabolism has worked for other nutrients such as phosphorus, nitrogen and rare earth metals, which promote primary production as well as the release of BGCs metabolites. For example, during actinorhodin biosynthesis by *Streptomyces lividans*, glucose supplementation was shown to down-regulate a known regulator of secondary metabolism (*AfsR*) [76].

Isolates are sequenced using whole genome sequencing and bypassing the culture-based screening to reduce numerous cultivation assays and avoid silencing novel BGCs under cultivation conditions, as well as prevent the time-consuming difficulties associated with conducting the primary and secondary screening of isolates. This approach was adopted to obtain about forty rare Actinomycetes from date palm rhizospheres in a saline environment, with all of the isolates being potential novel BGC producers [16].

The gene datasets across the genome-encoding specialized metabolic enzymes can be identified using the computer algorithms antiSMASH (for functional and comparative analysis) and PRISM (for the prediction of chemical structures that can be used to provide a pre-metabolomics insight on possible structure). Whole genome sequencing of three Actinomycetes revealed the presence of more BGCs using antiSMASH 5.0 and known clusterFINDER predictive tools. Hence, computational predictions could match several BGCs to known molecules with high-percentage similarities. In addition, organisms with more BGCs that are less similar to the gene sequence clusters are likely to experimentally produce unknown molecules and should be given priority. In this regard, genome sequencing of rare *Sacharothrix* genera with a predicted antiSMASH approach was used to match their gene clusters with non-ribosomal peptide metabolites with low similarity rates to known antibiotics [77].

The rate of productivity and agro-activity of the strain of interest may be enhanced through physical and chemical mutagenesis. For example, during the production of antimycin, an active insecticide purified using solvent partition, mid-pressure liquid chromatography, Sephadex LH20 column chromatography and high-performance liquid chromatography, physical mutagenesis was used to enhance its insecticidal activity. Ultraviolet irradiation mutagenic treatment of an Actinomycetes strain enhanced the production of antibiotic carbomycin with improved antibiotics activity compared to the wild-type. In addition, single and double exposure of the *Streptomyces* sp. KR0006 strain to ultraviolet irradiation revealed improvements of 70% and 100%, respectively, in strain insecticidal activity. Therefore, well-guided isolation, screening, characterization, taxonomic identification, whole genome sequencing with computational predictions, fermentation and metabolomics could reveal new agro-antibiotics applicable for plant protection. In ad-

dition, the strain activity can be improved through mutagenesis, protoplast fusion and metabolic regulations.

## 8. Conclusions and Future Prospects

Rare Actinomycetes genera appear to be a treasure trove for the biodiscovery of several compounds applicable in ensuring plant health through their agro-activity potential, which is a function of the agro-antibiotics they release both in planta and in vitro. They have highly conserved gene clusters from which novel antibiotics can be expressed, as long as experimental conditions are well guided to activate novel BGCs with the potential to biosynthesize novel compounds. Culture-dependent isolation and genometabolomic systems are highly promising in the production of agroactive compounds, although most scientists opt for the cost-ineffective metagenomics routes, while whole genome sequencing with the right computational predictive tools can reveal novel metabolites that can be purified and elucidated.

In a rapid targeted agro-antibiotics discovery, genome engineering using CRISPR-Cas9 in combination with homologous or heterologous expression of genes in new chassis from which endogenous genes have been removed to avoid confounding effects, as well as a targeted pathway scaffold exposé using tailored chassis through synthetic biology, can help to produce conserved silent BGCs with unknown scaffold elucidated/unveiled, making unknown agro-antibiotics known on a larger scale. In addition, more derivatives and analogues can be discovered, hence developing the production of biopesticides for food security assurance.

**Author Contributions:** O.P.O. was responsible for writing the review, while O.O.B. was responsible for supervision, critical revision of various drafts, quality assurance and funding acquisition. While B.R.G., D.D., W.Y. and G.S. critically reviewed the final draft.. All authors have read and agreed to the published version of the manuscript.

**Funding:** This study was funded by the National Research Foundation (NRF) of South Africa, grants (UID: 123634; UID132595; O.O.B.), while the tuition and stipend of O.P.O. was funded by NRF, grant UID: 138583.

**Acknowledgments:** O.O.B. appreciates the NRF of South Africa for the grants (UID: 123634; UID132595) that have supported research in her laboratory. O.P.O. would like to thank NRF for the support (UID: 138583) and North-West University for the bursary. We appreciate B.R.G., D.D., W.Y. and G.S. for their intellectual comments on the manuscript.

**Conflicts of Interest:** The authors report no declarations of interest.

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
