# Peer review of "Sustainable Agriculture: Rare-Actinomycetes to the Rescue"

_agronomy, doi:10.3390/agronomy13030666_

Round 1
Reviewer 1 Report
Summary,
The authors have discussed sustainable and agricultural intensification in saving ecosystems or public health using actinomycetes in this review article. However, rather than focusing on the usage of actinomycetes or their products to improve the ecosystem and public health, they have focused on sustainable agriculture and its history. Generally, they have tried to amalgamate sustainable agriculture with actinomycetes however sustainable agriculture is not solely about actinomycetes usage, it is a broad terminology where scientists, stakeholders, communities, and beneficial microbes all play their roles. This review article can be improved by focusing on the role of actinomycetes in sustainable agriculture rather than focusing on sustainable agriculture. Following are major comments and suggestions;
Lines 16-26; Good story about the conservation of the ecosystem and protection of public health using sustainable agriculture but not good review lines. Not a single line about the actinomycetes, therefore there is a need to rewrite the abstract.
Lines 36-131: There are many stories, from the UN vision to climate change to actinomycetes. What do you want to introduce? Why do you want to introduce yourself? Why do you want to write this review? What are the objectives of the review? I suggest, focusing on actinomycetes and their role in sustainable agriculture rather than writing about climate change, visions, etc.
Figure 1: What is this legend {Smart Isolation Roadmap}. What is isolation? Redraw the figure and mention parts of the figure with different letters. Write legend properly? Not a good figure.
Lines 135-139: History of sustainable agricultural practices? I suggest writing this section again with the title, history of actinomycetes’ and their role in sustainable agriculture practices, I mean focus on actinomycetes?
Section 4: 229-280: Rare Actinomycetes to the rescue: Why only rare actinomycetes? Why are they rare? What about the roles of their genes and their antibiotics? Elaborate this section with examples of biopesticides used against pests and plants diseases and antibiotics against human diseases, So human and plant health, and wellness by using actinomycetes will be a great story. Also, how do actinomycetes improve the environment, discuss one portion of this topic.
Figure 2: Is not of high quality, look legends: Figure 2. Characteristics of Agricultural Systems Applied for Ensuring Disease Resilience, many capital letters within the sentence, please construct informative figures.
Discuss genomics, transcriptomics, and specially metabolomic studies on the actinomycetes,
Draw table/tables, with column headings, actinomycetes species/strains, gene/genes, their products, usages, against diseases, of plants, humans, and against pests, with references.
Author Response
Line 16-23-Good story about the conservation of the ecosystem and protection of public health using sustainable agriculture but not good review lines. Not a single line about the actinomycetes, therefore there is a need to rewrite the abstract.
Response: I have modified the abstract.
Lines 36-131: There are many stories, from the UN vision to climate change to actinomycetes. What do you want to introduce? Why do you want to introduce yourself? Why do you want to write this review? What are the objectives of the review? I suggest, focusing on actinomycetes and their role in sustainable agriculture rather than writing about climate change, visions, etc.
Response 2: The reason for writing the review has been stated.
Figure 1: What is this legend {Smart Isolation Roadmap}. What is isolation? Redraw the figure and mention parts of the figure with different letters. Write legend properly? Not a good figure.
Response: The figure has been removed.Lines 135-139: History of sustainable agricultural practices? I suggest writing this section again with the title, history of actinomycetes’ and their role in sustainable agriculture practices, I mean focus on actinomycetes?
Response 3: Corrected.
Section 4: 229-280: Rare Actinomycetes to the rescue: Why only rare actinomycetes? Why are they rare? What about the roles of their genes and their antibiotics? Elaborate this section with examples of biopesticides used against pests and plants diseases and antibiotics against human diseases, So human and plant health, and wellness by using actinomycetes will be a great story. Also, how do actinomycetes improve the environment, discuss one portion of this topic.
Response 4: Rare Actinomycetes is the focus for my program, hence i tried to knock out Streptomycetes.
Figure 2: Is not of high quality, look legends: Figure 2. Characteristics of Agricultural Systems Applied for Ensuring Disease Resilience, many capital letters within the sentence, please construct informative figures.
Response: Upgraded.
Discuss genomics, transcriptomics, and specially metabolomic studies on the actinomycetes,
Response: Well, i tried to save an indept review on all subjects for the next review. But i had given an overview.
Draw table/tables, with column headings, actinomycetes species/strains, gene/genes, their products, usages, against diseases, of plants, humans, and against pests, with references.
Response: added this based on scope.
Reviewer 2 Report
The review by Oyedih et al. provides a comprehensive overview of sustainable agriculture, historical obstacles during the history, and rare Actinomycetes as a source of potential new biopesticides. In order to provide a clear overview of the known metabolites, a table summarizing the important bioactive metabolites produced by rare Actinomycetes should be prepared. Some technical comments and suggestions are also provided below:
Line 88: change "produce genes coding" to "that their genomes are coding"
Line 89: delete "another plant"
Line 95: change "subsets" to "genera"
Line 101: it is better to use the term "biopesticide" instead of "agro-antibiotic" because antibiotic is often referred to as antibacterial and biopesticide has a broader meaning
Line 133: Figure 1 needs to be presented in a better resolution
Line 173: the abbreviation for sustainable intensification has already been introduced, so just leave S.I. in the sentence
Line 239: introduce here that these genera are rare Actinomycetes
Line 258: change "microm" to "µm"
Line 260-262: this sentence should be omitted, it is too general
Line 263: delete “which are encoded in biosynthetic genes”
Line 273: remove hyphen from in vivo and in planta
Line 299: change "large genome makeup (G + C content)" to "have high GC content"
Line 315: change "worms" to "roundworms"
Line 317: list at the end of the sentence before explaining each in the next subsections
Line 356: remove parentheses around chitinase
Line 396: insert "bacterium" before "Pseudomonas syringae"
Line 436: did you mean septum?
Line 446: considering that the main objective of the review is metabolites of rare Actinomycetes, Streptomyces should be omitted. Revise this subsection
Line 458: change "production" to "mining"
Line 483: Streptomyces again, focus only on rare Actinomycetes
Line 531: change "in plant" to "in planta"
Line 532: change "released" to "expressed"
Author Response
Point 1: In order to provide a clear overview of the known metabolites, a table summarizing the important bioactive metabolites produced by rare Actinomycetes should be prepared.
Response 1: This review focuses on sustainable agriculture and Actinomycetes as a potential source of agro-antibiotics, I have a review on metabolites already.
Point 2: Some technical comments and suggestions are also provided below:
Response 2: All technical comments have been addressed, except that for line 101, because in the agricultural sector, agro-antibiotics is a term applicable when dealing with bacterial and fungal phytopathogens.